# VER: Scaling On-Policy RL Leads to the Emergence of Navigation in Embodied Rearrangement

**Erik Wijmans**[1,2]    **Irfan Essa**[1,3]    **Dhruv Batra**[2,1]
[1] Georgia Institute of Technology    [2] Meta AI    [3] Google Atlanta
{etw,irfan,dbatra}@gatech.edu

## Abstract

We present Variable Experience Rollout (VER), a technique for efficiently scaling batched on-policy reinforcement learning in heterogenous environments (where different environments take vastly different times to generate rollouts) to many GPUs residing on, potentially, many machines. VER combines the strengths of and blurs the line between synchronous and asynchronous on-policy RL methods (SyncOnRL and AsyncOnRL, respectively). Specifically, it learns from on-policy experience (like SyncOnRL) and has no synchronization points (like AsyncOnRL) enabling high throughput.

We find that VER leads to significant and consistent speed-ups across a broad range of embodied navigation and mobile manipulation tasks in photorealistic 3D simulation environments. Specifically, for PointGoal navigation and ObjectGoal navigation in Habitat 1.0, VER is 60-100% faster (1.6-2x speedup) than DD-PPO, the current state of art for distributed SyncOnRL, with similar sample efficiency. For mobile manipulation tasks (open fridge/cabinet, pick/place objects) in Habitat 2.0 VER is 150% faster (2.5x speedup) on 1 GPU and 170% faster (2.7x speedup) on 8 GPUs than DD-PPO. Compared to SampleFactory (the current state-of-the-art AsyncOnRL), VER matches its speed on 1 GPU, and is 70% faster (1.7x speedup) on 8 GPUs with better sample efficiency.

We leverage these speed-ups to train chained skills for GeometricGoal rearrangement tasks in the Home Assistant Benchmark (HAB). We find a surprising *emergence of navigation* in skills that do not ostensible require any navigation. Specifically, the Pick skill involves a robot picking an object from a table. During training the robot was always spawned close to the table and never needed to navigate. However, we find that if base movement is part of the action space, the robot learns to navigate *then* pick an object in new environments with 50% success, demonstrating surprisingly high out-of-distribution generalization.

Code: github.com/facebookresearch/habitat-lab

## 1   Introduction

Scaling matters. Progress towards building embodied intelligent agents that are capable of performing goal driven tasks has been driven, in part, by training large neural networks in photo-realistic 3D environments with deep reinforcement learning (RL) for (up to) billions of steps of experience [Wijmans et al., 2020, Maksymets et al., 2021, Mezghani et al., 2021, Ramakrishnan et al., 2021, Miki et al., 2022]. To enable this scale, RL systems must be able to efficiently utilize the available resources (*e.g.* GPUs), and scale to multiple machines all while maintaining sample-efficient learning.

One promising class of techniques to achieve this scale is batched on-policy RL. These methods collect experience from many ($N$) environments simultaneously using the policy and update it with this cumulative experience. They are broadly divided into two classes: synchronous (SyncOnRL)

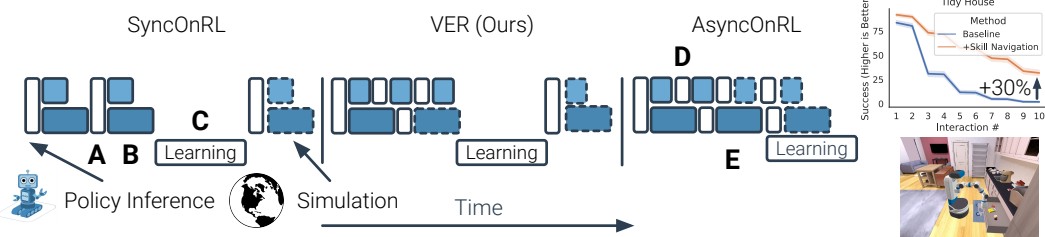

Figure 1: **(Left) RL Training Systems.** In SyncOnRL, actions are computed for all environments, then all environments are stepped. Experience collection is paused during learning. In AsyncOnRL, computing actions, stepping environments, and learning all occur without synchronization. In VER, a variable amount of experience is collected from each environment, enabling synchronous learning without the straggler effect. **(Right) skill policies** with navigation are more robust to handoff errors.

and asynchronous (AsyncOnRL). SyncOnRL contains two synchronization points: first the policy is executed for the entire batch $(o_t \rightarrow a_t)_{b=1}^{B}$ [1] (Fig. 1 A), then actions are executed in *all* environments, $(s_t, a_t \rightarrow s_{t+1}, o_{t+1})_{b=1}^{B}$ (Fig. 1 B), until $T$ steps have been collected from all $N$ environments. This $(T, N)$-shaped batch of experience is used to update the policy (Fig. 1 C). Synchronization reduces throughput as the system spends significant (sometimes the most) time waiting for the slowest environment to finish. This is the straggler effect [Petrini et al., 2003, Dean and Ghemawat, 2004].

AsyncOnRL removes these synchronization points, thereby mitigating the straggler effect and improving throughput. Actions are taken as soon as they are computed, $a_t \rightarrow o_{t+1}$ (Fig. 1 D), the next action is computed as soon as the observation is ready, $o_t \rightarrow a_t$ (Fig. 1 E), and the policy is updated as soon as enough experience is collected. However, AsyncOnRL systems are not able to ensure that all experience has been collected by only the current policy and thus must consume *near*-policy data. This reduces sample efficiency [Liu et al., 2020]. Thus, status quo leaves us with an unpleasant tradeoff – high sample-efficiency with low throughput or high throughput with low sample-efficiency.

In this work, we propose Variable Experience Rollout (VER). VER combines the strengths of and blurs the line between SyncOnRL and AsyncOnRL. Like SyncOnRL, VER collects experience with the current policy and then updates it. Like AsyncOnRL, VER does not have synchronization points – it computes next actions, steps environments, and updates the policy as soon as possible. The inspiration for VER comes from two key observations:

1) AsyncOnRL mitigates the straggler effect by implicitly collecting a variable amount of experience from each environment – more from fast-to-simulate environments and less from slow ones.

2) Both SyncOnRL and AsyncOnRL use a fixed rollout length, $T$ steps of experience. Our key insight is that while a fixed rollout length may simplify an implementation, it is *not* a requirement for RL.

These two key observations naturally lead us to *variable experience rollout* (VER), *i.e.* collecting rollouts with a variable number of steps. VER adjusts the rollout length for each environment based on its simulation speed. It explicitly collects more experience from fast-to-simulate environments and less from slow ones (Fig. 1). The result is an RL system that overcomes the straggler effect *and* maintains sample-efficiency by learning from on-policy data.

VER focuses on efficiently utilizing a single GPU. To enable efficient scaling to multiple GPUs, we combine VER with the decentralized distributed method proposed in [Wijmans et al., 2020].

First, we evaluate VER on well-established embodied navigation tasks using Habitat 1.0 [Savva et al., 2019] on 8 GPUs. VER trains PointGoal navigation [Anderson et al., 2018] 60% faster than Decentralized Distributed PPO (DD-PPO) [Wijmans et al., 2020], the current state-of-the-art for distributed on-policy RL, with the same sample efficiency. For ObjectGoal navigation [Batra et al., 2020b], an active area of research, VER is 100% faster than DD-PPO with (slightly) better sample efficiency.

Next, we evaluate VER on the recently introduced (and significantly more challenging) GeometricGoal rearrangement tasks [Batra et al., 2020a] in Habitat 2.0 [Szot et al., 2021]. In GeoRearrange, a virtual robot is spawned in a new environment and asked to rearrange a set of objects from their initial to

---

[1]Following standard notation, $s_t$ is (PO)MDP state, $a_t$ is the action taken, and $o_t$ is the agent observation.

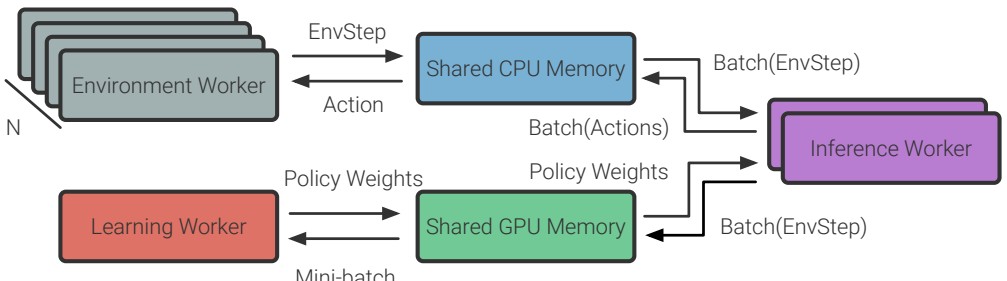

Figure 2: **VER system architecture.** Environment workers receive actions to simulate and return the result of that environment step (EnvStep). Inference workers receive batches experience from environment workers. They return the new action to take to environment workers and write the experience into GPU shared memory for learning.

desired coordinates. These environments have highly variable simulation time (physics simulation time increases if the robot bumps into something) and require GPU-acceleration (for photo-realistic rendering), limiting the number of environments that can be run in parallel.

On 1 GPU, VER is 150% faster (2.5x speedup) than DD-PPO with the same sample efficiency. VER is as fast as SampleFactory [Petrenko et al., 2020], the state-of-the-art AsyncOnRL, with the same sample efficiency. VER is as fast as AsyncOnRL in pure throughput; this is a surprisingly strong result. AsyncOnRL never stops collecting experience and should, in theory, be a strict upper bound on performance. VER is able to match AsyncOnRL for environments that heavily utilize the GPU for rendering, like Habitat. In AsyncOnRL, learning, inference, and rendering contend the GPU which reduces throughput. In VER, inference and rendering contend for the GPU while learning does not.

On 8 GPUs, VER achieves better scaling than DD-PPO, achieving a 6.7x speed-up (vs. 6x for DD-PPO) due to lower variance in experience collection time between GPU-workers. Due to this efficient multi-GPU scaling, VER is 70% faster (1.7x speedup) than SampleFactory on 8 GPUs and has better sample efficiency as it learns from on-policy data.

Finally, we leverage these SysML contributions to study open research questions posed in prior work. Specifically, we train RL policies for mobile manipulation skills (Navigate, Pick, Place, *etc.*) and chain them via a task planner. Szot et al. [2021] called this approach TP-SRL and identified a critical 'handoff problem' – downstream skills are set up for failure by small errors made by upstream skills (*e.g.* the Pick skill failing because the navigation skill stopped the robot a bit too far from the object).

We demonstrate a number of surprising findings when TP-SRL is scaled via VER. Most importantly, we find the *emergence of navigation* when skills that do not ostensibly require navigation (*e.g.* pick) are trained with navigation actions enabled. In principle, Pick and Place policies do not *need* to navigate during training since the objects are always in arm's reach, but in practice they learn to navigate to recover from their mistakes and this results in strong out-of-distribution test-time generalization. Specifically, TP-SRL *without a navigation skill* achieves 50% success on NavPick and 20% success on a NavPickNavPlace task simply because the Pick and Place skills have learned to navigate (sometimes across the room!). TP-SRL with a Navigate skill performs even stronger: 90% on NavPickNavPlace and 32% on 5 successive NavPickNavPlaces (called Tidy House in Szot et al. [2021]), which are +32% and +30% absolute improvements over Szot et al. [2021], respectively. Prepare Groceries and Set Table, which both require interaction with articulated receptacles (fridge, drawer), remain as open problems (5% and 0% Success, respectively) and are the next frontiers.

## 2   VER: Variable Experience Rollout

The key challenge that any batched on-policy RL technique needs to address is variability of simulation time for the environments in a batch. There are two primary sources of this variability: action-level and episode-level. The amount of time needed to simulate an action within an environment varies depending on the the specific action, the state of the robot, and the environment (*e.g.* simulating the robot navigating on a clear floor is much faster than simulating the robot's arm colliding with objects). The amount of time needed to simulate an entire episode also varies environment to environment irrespective of action-level variability (*e.g.* rendering images takes longer for visually-complex scenes, simulating physics takes longer for scenes with a large number of objects).

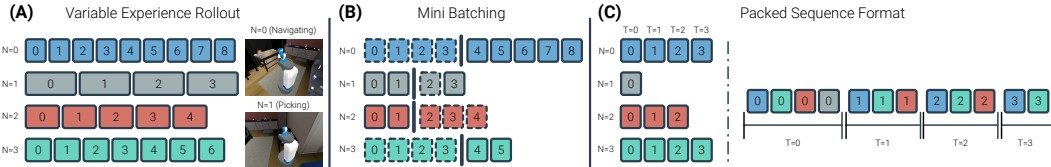

Figure 3: **(A)** `VER` collects a variable amount of experience from each environment. The length of each step represents the time taken to collect it. **(B)** `VER` **mini-batch.** The solid bars denote episode boundaries. The steps selected for the first mini-batch have a dashed border. **(C)** **The** `PackedSequence` **data format** represents a set of sequences with variable length in a linear buffer such that all elements from each timestep area next to one-another in memory.

## 2.1 Action-Level Straggler Mitigation

We mitigate the action-level straggler effect by applying the experience collection method of `AsyncOnRL` to `SyncOnRL`. We represent this visually in Fig. 2 and describe it in text bellow.

**Environment workers** receive the next action and step the environment, EnvStep, *e.g.* $s_t, a_t \rightarrow s_{t+1}, o_{t+1}, r_t$. They write the outputs of the environment (observations, reward, *etc.*) into pre-allocated CPU shared memory for consumption by inference workers.

**Inference workers** receive batches of steps of experience from environment workers. They perform inference with the current policy to select the next action and send it to the environment worker using pre-allocated CPU shared memory. After inference they store experience for learning in shared GPU memory. Inference workers use dynamic batching and perform inference on all outstanding inference requests instead of waiting for a fixed number of requests to arrive[2]. This allows us to leverage the benefits of batching without introducing synchronization points between environment workers.

This experience collection technique is similar to that of HTS-RL [Liu et al., 2020] (`SyncOnRL`) and SampleFactory [Petrenko et al., 2020] (`AsyncOnRL`). Unlike both, we do not overlap experience collection with learning. This has various system benefits, including reducing GPU memory usage and reducing GPU driver contention. More details are available in Appendix A.

## 2.2 Environment-Level Straggler Mitigation

In both `SyncOnRL` and `AsyncOnRL`, the data used for learning consists of $N$ rollouts of equal-$T$ steps of experience, an $(T, N)$-shaped batch. In `SyncOnRL` these $N$ sets are all collected with the current policy, this leads to the environment-level straggler effect. `AsyncOnRL` mitigates this by relaxing the constraint that experience must be strictly on-policy, and thereby implicitly changes the experience collection rate for each environment.

**Variable Experience Rollout (`VER`).** We instead relax the constraint that we must use $N$ rollouts of equal-$T$ steps. Specifically, `VER` collects $T \times N$ steps of experience from $N$ environments without a constraint on how many steps of experience are collected from each environment. This explicitly varies the experience collection rate for each environment – in effect, collecting more experience from environments that are fast to simulate. Consider the 4 environments shows in Fig. 3A. The length of the each step representation the wall-clock time taken to collect it, some steps are fast, some are slow. `VER` collects more experience from environment 0 as it is fastest to step and less from 1, the slowest.

**Learning mini-batch creation.** `VER` is designed with recurrent policies in mind because memory is key in long-range and partially observable tasks like HAB. When training recurrent policies, we must create mini-batches of experience with sequences for back-propagation-through-time. Normally $B$ mini-batches are constructed by spitting the $N$ environments' experience into $B$ $(T, {}^N\!/_B)$-sized mini-batches. A similar procedure would result in mini-batches of different sizes with `VER`. This would harm optimization because learning rate and optimization mini-batch size are intertwined and automatically adjusting the learning rate is an open question [Goyal et al., 2017, You et al., 2020].

To under `VER`'s mini-batching, first note that there are two reasons for the start of a new sequence of experience: rollout starts (Fig. 3B, step 0) and episode starts (Fig. 3B, a step after a bar). These

---

[2]In practice we introduce both a minimum and maximum number of requests to prevent under-utilization of compute and over-utilization of memory.

two boundaries types are independent – episodes can end at any arbitrary step within the rollout and then that environment will reset and start a new episode. Thus when we collect experience from $N$ environments, we will have $K \geq N$ sequences to divide between the mini-batches. We distributed these $K$ sequences between the mini-batches. We randomly order the sequences, then the first $T \times N/B$ steps are the first mini-batch, the next $T \times N/B$ to the second, *etc.* See Fig. 3B for an example.

**Batching computation for learning.** The mini-batches constructed from the algorithm above have sequences with variable length. To batch the computation of these sequences we use cuDNN's PackedSequence data model. This data model represents a set of variable-length sequences (Fig. 3C left) such that all sequence-elements at a time-step are contiguous in memory (Fig. 3C right) – enabling batched computation on each time-step for components with a temporal dependence, *e.g.* the RNN – and that all elements across all time-steps are also contiguous – enabling batched computation across *all* time-steps for network components that don't have a temporal dependence, *e.g.* the visual encoder.

During experience collection we write experience into a linear buffer in GPU memory and then arrange each mini-batch as a PackedSequence. This takes less than 10 milliseconds (per learning phase); orders of magnitude less than experience collection ($\sim$3s) or learning ($\sim$1.5s) in our experiments.

**Learning method.** The experience and mini-batches generated from VER are well-suited for use with RL methods that use on-policy data [Sutton and Barto, 1992]. We use Proximal Policy Optimization (PPO) [Schulman et al., 2017] as it is known to work well for embodied AI tasks and recurrent policies.

**Inflight actions** One subtle design choice is the following – when VER finishes a $T \times N$ experience collection, there will be (slow) environments that haven't completed simulation yet. Instead of discarding that data, we choose to collect this experience in the *next* rollout. We find this choice leads to speed gains without any sample-efficiency loss.

## 2.3  Multiple GPUs

We leverage the decentralized distributed training architecture from Wijmans et al. [2020] to scale VER to multiple GPUs (residing on a single or multiple nodes). In this architecture, each GPU both collects experience and learns from that experience. During learning, gradients from each GPU-worker are averaged with an AllReduce operation. This is a synchronous operation and thus introduces a GPU-worker-level straggler effect. Wijmans et al. [2020] mitigate this effect by preempting stagglers – stopping collecting experience early and proceeding to learning – after a fixed number of GPU-workers have finished experience collection.

We instead approximate the optimal preemption for each experience collection phase. Given the learning time, LT, and a function $\text{Time}(S)$ that returns the time needed to collect $S$ steps of experience, the optimal number of steps $S$ to collect before preempting stragglers is $\max_S {}^S/_{(\text{Time}(S)+\text{LT})}$, s.t. $S \leq (T \cdot N \cdot \text{\#GPUs})$. For LT, we record the time from the last iteration as this doesn't change between iterations. We estimate $\text{Time}(S)$ using the average step time to receive a step of experience from each environment (this includes both inference time and simulation time) from the previous rollout. This will be a poor estimation in some circumstances but we find it to work well in practice.

We improve upon sample efficiency of DD-PPO by filling the preempted rollouts with experience from the previous rollout. We perform extra epochs of PPO on this now 'stale' data instead of correcting for the off-policy return as we find this simpler and effective. This comes with no effective computation cost since the maximum batch size per GPU is unchanged.

## 3  Embodied Navigation: Benchmarking

First, we benchmark VER on the embodied navigation [Anderson et al., 2018] tasks in Habitat 1.0 [Savva et al., 2019] – PointNav [Anderson et al., 2018] and ObjectNav [Batra et al., 2020b]. Our goal here is simply to show training speed-ups in well-studied tasks (and in the case of PointNav, a well-saturated task with no room left for accuracy improvements). We present accuracy improvements and in-depth analysis on challenging rearrangement tasks in Section 6.

For both tasks, we use standard architectures from Habitat Baselines [Savva et al., 2019, Wijmans et al., 2020] – the ResNet18 encoder and a 2 layer LSTM. Following Ye et al. [2020, 2021], we add Action Conditional Contrastive Coding [Guo et al., 2018].

**PointNav.** We train PointNav agents with one RGB camera on the HM3D dataset [Ramakrishnan et al., 2021] for 1.85 billion steps of experience on 8 GPUs. We study the RGB setting (and not Depth) because this is the more challenging version of the task and thus we expect it to be more sensitive to

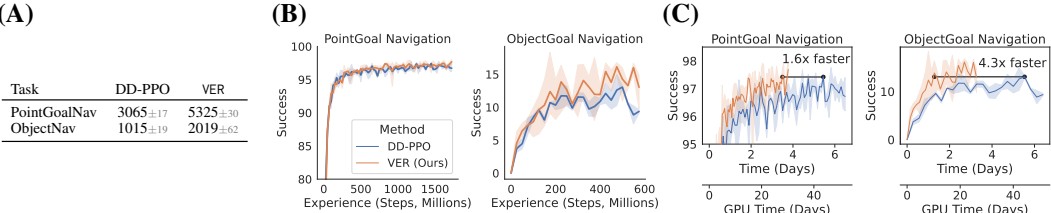

Figure 4: **(A) Navigation Tasks** training steps per second on 8 GPUs. VER is 60%-100% faster than DD-PPO. **(B) Sample efficiency** on ObjectNav and PointNav performance (validation success). VER has similar or slightly better sample efficiency than DD-PPO, indicating that performance is not negatively impacted by the non-uniform sampling of experience from environments. Shading is a 95% confidence interval over 3 seeds. **(C) Compute efficiency** on ObjectNav and PointNav performance (validation success). To reach the maximum success achieved by DD-PPO, 97.4% on PointNav and 13.0% on ObjectNav, VER uses 1.6x less compute on PointNav (saving 16 GPU-days) and 4.3x less compute on ObjectNav (saving 33.4 GPU-days).

possible differences in the training system. We examine VER along two axes: 1) training throughput – the number of samples of experience per second (SPS) the system collects and learns from, 2) sample efficiency. On 8 GPUs, VER trains agents 60% faster than DD-PPO, from 3065 SPS to 5325 SPS (Fig. 4A), with similar sample efficiency (Fig. 4B).

**ObjectNav.** We train ObjectNav agents with one `RGB` and one `Depth` camera on the MP3D [Chang et al., 2017] dataset for 600 million steps of experience on 8 GPUs. VER trains agents 100% faster than DD-PPO, 1021 to 2019 (Fig. 4A), with slightly better sample efficiency (Fig. 4B). Due to improved throughput and sample efficiency, VER uses 4.3x less compute than DD-PPO to reach the same success (Fig. 4C). There are two effects that enable better sample efficiency with VER. First, we perform additional epochs of PPO on experience from the last rollout when a GPU-worker is preempted. Second, the variable experience rollout mechanism results in a natural curriculum. Environments are often faster to simulate when they are easier (*i.e.* a smaller home), so more experience will be collected in these easier cases.

## 4 Embodied Rearrangement: Task, Agent, and Training

Next, we use VER to study the recently introduced (and more challenging) GeometricGoal rearrangement rearrangement tasks [Batra et al., 2020a] in Habitat 2.0 [Szot et al., 2021].

**Task.** In GeoRearrange, an agent is initialized in an unknown environment and tasked with rearranging objects in its environment. The task is specified as a set of coordinate pairs $\{(\text{Pose}_{\text{Initial}}, \text{Pose}_{\text{Final}})\}_{o=1}^{O}$. The agent must bring each object at $\text{Pose}_{\text{Initial}}$ to $\text{Pose}_{\text{Final}}$ where Pose is the initial or desired center-of-mass location for the object(s). We use the Home Assistant Benchmark (HAB) which consists of 3 scenarios of increasing difficulty: Tidy House, Prepare Groceries, and Set Table.

In Tidy House, the agent is tasked with moving 5 objects from their initial locations to their final locations. The objects are rarely in containers (*i.e.* fridge or cabinet drawer) and when they are, the containers are already opened. In Prepare Groceries, the agent must move 3 objects from the kitchen counter into the open fridge (or open fridge to counter). This stresses picking and placing in the fridge, which is challenging. In Set Table, the agent must move 1 object from the closed kitchen cabinet drawer to the table and 1 object from the closed fridge to the table. This requires opening the cabinet and fridge, and picking from these challenging receptacles.

**Simulation.** We use the Habitat simulator with the ReplicaCAD Dataset [Savva et al., 2019, Szot et al., 2021]. The robot policy operates at 30 Hz and physics is simulated at 120 Hz.

**Agent.** The agent is embodied as a Fetch robot with a 7-DOF arm. The arm is controlled via joint velocities. At every time step the policy predicts a delta in motor position for joint in the arm. We find joint velocity control equally easy to learn but faster to simulate than the end-effector control used in Szot et al. [2021]. The arm is equipped with a suction gripper. The agent must control the arm such that the gripper is in contact with the object to grasp and then activate the gripper. The object is dropped once the gripper is deactivated. This is more realistic than the 'magic' grasp action used in Szot et al. [2021]. The robot base (navigation) is controlled by the policy commanding a desired linear speed and

| | DD-PPO | | NoVER | | VER (Ours) | | SampleFactory | |
|---|---|---|---|---|---|---|---|---|
| GPUs | Mean | Max | Mean | Max | Mean | Max | Mean | Max |
| 1 | $174_{\pm7}$ | $442_{\pm9}$ | $327_{\pm7}$ | $428_{\pm11}$ | $428_{\pm5}$ | $534_{\pm7}$ | $427_{\pm5}$ | $517_{\pm3}$ |
| 2 | $283_{\pm23}$ | $696_{\pm24}$ | $592_{\pm5}$ | $786_{\pm11}$ | $716_{\pm32}$ | $945_{\pm29}$ | $804_{\pm15}$ | $1022_{\pm0}$ |
| 4 | $468_{\pm21}$ | $1337_{\pm34}$ | $1097_{\pm30}$ | $1601_{\pm39}$ | $1432_{\pm10}$ | $1915_{\pm13}$ | $1286_{\pm6}$ | $1568_{\pm26}$ |
| 8 | $1066_{\pm84}$ | $2754_{\pm156}$ | $2216_{\pm60}$ | $3438_{\pm94}$ | $2861_{\pm21}$ | $3829_{\pm23}$ | $1662_{\pm4}$ | $1842_{\pm0}$ |

Table 1: **SyncOnRL, VER, and AsyncOnRL benchmarking.** Mean/max system throughput (SPS) over 20 million training steps. VER is 150% faster than DD-PPO on 1 GPU and 170% faster on 8 GPUs. Hardware: Tesla V100(s) with 10 CPUs per GPU.

angular velocity. The robot is equipped with one Depth camera attached to its head, proprioceptive sensors that provide the joint positions of its arm, and a GPS+Compass sensor that provides its heading and location relative to its initial location. The policy models $a_t \sim \pi(\cdot \mid s_{t-1})$ instead of $a_t \sim \pi(\cdot \mid s_t)$. This is more realistic and enables physics and rendering to be overlapped [Szot et al., 2021].

We build upon the TaskPlanning-SkillRL (TP-SRL) method proposed in Szot et al. [2021]. TP-SRL is a hierarchical method for GeoRearrange that decomposes the task into a series of skills – Navigate, Pick, Place, and {Open, Close } × {Cabinet, Fridge}. Skills are controlled via a skill-policy (learned with RL) and chained together via a task planner. One of the key challenges is the 'handoff problem' – downstream skills are setup for failure due to slight errors made by the upstream skill. We give all skill policies access to navigation actions to allow them to correct for these errors.

**Architecture.** All skill policies share the same architecture. We use ResNet18 [He et al., 2016] to process the $128{\times}128$ visual input. Following Wijmans et al. [2020], we reduce with width of the network by half and use GroupNorm [Wu and He, 2018]. We also apply some of the recent advancements from ConvNeXt [Liu et al., 2022]. We use a patch-ify stem, dedicated down-sample stages, layer scale [Touvron et al., 2021], and dilated convolutions [Yu and Koltun, 2015] (this mimics larger kernel convolutions without increasing computation). The visual embedding is then combined with the proprioceptive observations and previous action, and then processed with a 2-layer LSTM [Hochreiter and Schmidhuber, 1997]. The output of the LSTM is used to predict the action distribution and value function. Actions are sampled from independent Gaussian distributions.

**Training.** We train agents using VER and PPO [Schulman et al., 2017] with Generalized Advantage Estimation [Schulman et al., 2016]. We use a minimum entropy constraint with a learned coefficient [Haarnoja et al., 2018] as we find this to be more stable given our diverse set of skills than a fixed coefficient. Formally, let $\mathcal{H}(\pi)$ be the entropy of the policy, we then minimize $\alpha([[\mathcal{H}(\pi)]]_{sg} - \lambda) - [[\alpha]]_{sg}\mathcal{H}(\pi)$ where $[[\cdot]]_{sg}$ is the stop gradient operator. We set the target entropy, $\lambda$, to zero for all tasks. We use the Adam optimizer [Kingma and Ba, 2015] with an initial learning rate of $2.5 \times 10^{-4}$ and decay it to zero with a cosine schedule. To correct for biased sampling in VER, we use truncated importance sampling weighting [Espeholt et al., 2018] with a maximum of 1.0.

## 5 Embodied Rearrangement: Benchmarking

In this section, we benchmark VER for training open-fridge policies because this task involves interaction of the robot with an articulated object (the fridge) and represents a challenging case for the training system due to large variability in physics time. In Section 6, we analyze the task performance, which requires all skills. For all systems, we set the number of environments, $N$, to 16 per GPU.

### 5.1 System throughout

**VER is 150% faster than DD-PPO** (Table 1); an even larger difference than in the simpler navigation tasks we studied before. DD-PPO has no mechanism to mitigate the action-level or episode-level straggler effects. In their absence, DD-PPO has similar throughput as VER (Table 1, Max). Under these effects, DD-PPO's throughput reduces 150% compared to 20% for VER (Table 1, Mean vs. Max).

**Variable experience rollouts are effective.** We compare VER with VER minus variable experience rollouts (NoVER). NoVER is a 'steel-manned' baseline for VER and benefits from all our micro-optimizations. VER is 30% faster than NoVER (Table 1).

**VER closes the gap to AsyncOnRL.** On 1 GPU, VER is as fast as SampleFactory [Petrenko et al., 2020], the fastest single machine AsyncOnRL. Intuitively AsyncOnRL should be a strict upper-bound on performance – it never stops collecting experience while VER does. However this doesn't take into

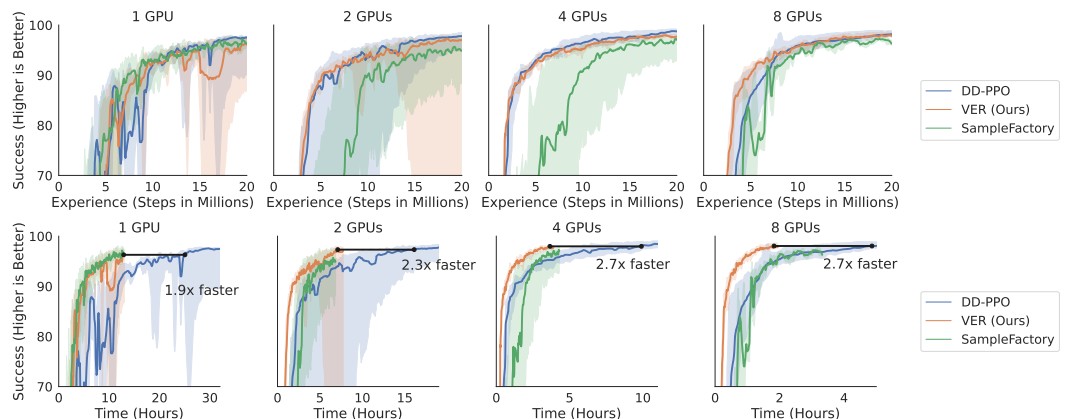

Figure 5: **Training efficiency** on Open Fridge. VER has similar sample efficiency as DD-PPO (SyncOnRL). SampleFactory (AsyncOnRL) has similar sample efficiency with 1 GPU but this reduces as policy lag increases with more GPUs. On 8 GPUs, VER uses 2.7x less compute than DD-PPO to reach the same performance. This saves 32-days of GPU-time, per skill, for the full training schedule of 500 million frames. The shaded region is a 95% bootstrapped confidence interval over 5 seeds. We use interquartile mean (IQM) as our summary statistic [Agarwal et al., 2021].

account the realities of hardware. Recall that we are training an agent with a large visual encoder. This means that updating the parameters of the agent takes a large amount of time ($\sim$150ms per mini-batch of size 1024 on a V100). Further, Habitat uses the GPU for rendering. The use of the GPU for both rendering and learning simultaneously results in GPU driver contention. In SampleFactory, learning time and experience collection time are roughly double that of VER.

**Multi-GPU scaling.** VER has better multi-GPU scaling than DD-PPO, achieving a 6.7x speed-up on 8 GPUs compared to 6x. The rollout collection time in VER is lower variance, which improves scaling. On 2 GPUs, SampleFactory is 12% faster than VER. Here one GPU is used for learning+inference and the other is used for rendering. This creates a nice division of work and doesn't result in costly GPU contention. On 4 and 8 GPUs however, the single GPU used for learning in SampleFactory is the bottleneck and VER has higher throughput (nearly 100% faster on 8 GPUs). While it is possible to implement multi-GPU learning for AsyncOnRL, it is left to the user to balance the number of GPUs used for experience collection and learning. VER balances GPU-time between these automatically.

## 5.2 Sample and Compute Efficiency

Next we examine sample efficiency of the training systems. On 1 GPU, VER has slightly worse sample efficiency than DD-PPO (Fig. 5). We suspect that this could be fixed by hyper-parameter choice. On 2 and 4 GPUs, VER has identical sample efficiency and better sample efficiency on 8 GPUs, Fig. 5 Interestingly, SampleFactory has similar sample efficiency with 1 GPU (Fig. 5) and is more stable. This is because reducing the number of GPUs also reduces the batch-size and PPO is known to not be batch-size invariant [Hilton et al., 2021]. We believe the stale data serves a similar function to PPO-EWMA [Hilton et al., 2021], which uses an exponential moving average of the policy weights for the trust region. On 2, 4, and 8 GPUs, VER has better sample efficiency than SampleFactory. Combing the findings of throughput and sample efficiency, we find that VER always achieves a given performance threshold in the least amount of training time (outright or a tie) (Fig. 5 Lower).

## 6 Embodied Rearrangement: Analysis of Learned Skills

We examine the performance of TP-SRL on the Home Assistant Benchmark (HAB) [Szot et al., 2021]. We examine both skill policies trained with the full action space and the limited, per-skill specific action space used in Szot et al. [2021]. Each skill is trained with VER for 500 million steps of experience on 8 GPUs. This takes less than 2 days per skill.

### 6.1 Performance on HAB

We find that skills with navigation actions improve performance on the full task but does not change performance on the skill's train task (*e.g.* Pick achieves 90% success with and without during training).

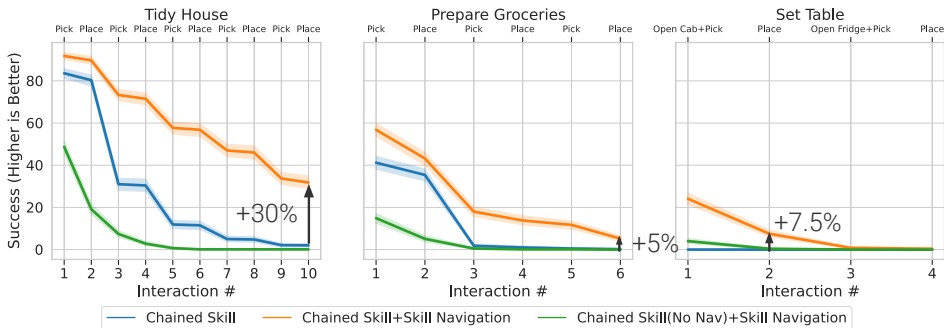

Figure 6: **HAB Performance** on the Tidy House, Prepare Groceries, and Set Table scenarios. Skill policies with navigation (TP-SRL+Skill Navigation) outperform skill policies without navigation (TP-SRL) despite not strictly needing this ability. Further, we find that these skill policies have learned *emergent* navigation. TP-SRL(NoNav)+Skill Navigation achieves 50% success on NavPick (Tidy House) and 20% success on NavPickNavPlace (Tidy House) despite all navigation being performed by skills policies that ostensibly don't need to navigate.

Fig. 6 shows smaller drops in performance between every interaction (there is a navigation between each interaction) demonstrating that skills with navigation effectively correct for handoff errors. This is impactful after place as the navigation policy tends to make more errors when navigating to the next location after place. On Tidy House, full task performance improves from 2% Success to 32%.

On Prepare Groceries (which requires picking/placing from/into the fridge) and Set Table (which requires opening the fridge/cabinet and then picking from it) performance improves slightly. 0% to 5% on Prepare Groceries and 0% to 7.5% on the first pick+place on Set Table. Both these tasks remain as open problems and the next frontier.

## 6.2 Emergent Navigation

Next we examine if the skill policies are able to navigate to correct for highly out-of-distribution initial location. We examine TP-SRL(NoNav), which that omits the navigation skill. In this agent *all* navigation is done by skill policies that ostensibly never needed to navigate during training.

We find *emergent navigation* in both the Pick and Place policies. TP-SRL(NoNav) achieves 50% success on NavPick (Fig. 6, Tidy House interaction 1), and 20% success on NavPickNavPlace (Fig. 6, Tidy House interaction 2). The latter is on-par with the TaskPlanning+SensePlanAct (TP-SPA) classical baseline and significantly better than the MonolithicRL baseline in Szot et al. [2021].

The Pick and Place policies were trained on tasks that requires no navigation but both are capable of navigation. We provided examples of both the training task for Pick and Place, and TP-SRL(NoNav) on Tidy House in the supplementary materials.

On Prepare Groceries and Set Table, the navigation performance of these policies is worse (-34% Success and -45% Success on the first interaction, respectively). Prepare Groceries requires picking from the fridge, which is challenging and requires navigation that doesn't accidentally close the door. Set Table requires opening the cabinet and then picking, which introduces an additional OpenCab skill and requires more precise navigation and picking from Pick. Performance is non-zero (15% and 4% on the first interaction, respectively) in both scenarios; indicting that the skill policies are capable of navigating even in these scenarios, albeit less successfully than in Tidy House.

We hypothesize that the Pick and Place polices learned navigation because this was useful *early* in training. Early in training the policies have yet to learn that only minimal navigation is needed to complete the task. Therefore the policy will sometimes cause itself to move away from the pick object/place location and will navigate back. Navigation is then not forgotten as the policy converges.

We examined the behavior of a Pick policy early in training and found that it does tend to move away from the object it needs to pick up and sometimes moves back. Although the magnitude of navigation is small and quite infrequent, so the degree of generalization is high.

This result, and higher performance on HAB, highlights that it may not always be beneficial to remove 'unneeded' actions. Szot et al. [2021] removed navigation where possible to improve sample efficiency

and training throughput[3]. Our experiments corroborate this; training without navigation improves both sample efficiency and training speed. However, by enabling navigation and allowing the agent to learn how to (not) use it, we arrived upon emergent navigation and improved HAB performance.

## 7 Related Work

`AsyncOnRL` methods provide high-throughput on-policy reinforcement learning [Espeholt et al., 2018, Petrenko et al., 2020]. However, they have reduced sample efficiency as they must correct for near-policy, or 'stale', data. Few support multi-GPU learning and, when they do, the user must manually balance compute between learning and experience collection [Espeholt et al., 2020]. VER achieves the same throughput on 1-GPU while learning with on-policy data, has better sample efficiency, supports multi-GPU learning, and automatically balances compute between learning and simulation.

`SyncOnRL`. HTS-RL [Liu et al., 2020] also use the experience collection techniques as `AsyncOnRL` to mitigate the action-level straggler effect. Unfortunately inefficiencies in the provided implementation prevent a meaningful direct comparison and hide the full effectiveness of this technique. In Appendix E we show that our re-implementation is 110% faster (2.1x speedup) and thus instead compare to the stronger baseline of NoVER in the main text. We propose a novel mechanism, variable experience rollouts, to mitigate the episode-level straggler effect and thereby close the gap to `AsyncOnRL`. We use and build upon Decentralized Distributed PPO (DD-PPO) [Wijmans et al., 2020], which proposed a distributed multi-GPU method based on data parallelism [Hillis and Steele Jr, 1986].

**Batched simulators** simulate multiple agents (in multiple environments) simultaneously and are responsible for their own parallelization [Shacklett et al., 2021, Petrenko et al., 2021, Freeman et al., 2021, Makoviychuk et al., 2021]. While these systems offer impressive performance, none currently support a benchmark like HAB (which combines physics and photo-realism) nor the flexibility of Habitat, AI2Thor [Kolve et al., 2017], or ThreeDWorld [Gan et al., 2020]. VER enables researchers to first explore promising directions using existing simulators and then build batched simulators with the knowledged gained from their findings.

## 8 Societal Impact, Limitations, and Conclusion

Our main application result is trained using the ReplicaCAD dataset [Szot et al., 2021], which is limited to only US apartments, and this may have negative societal impacts for deployed assistants. VER was designed and evaluated for tasks with both GPU simulation and large neural networks. For tasks with CPU simulation and smaller networks, we expect it to improve upon `SyncOnRL` but it may have less throughput than `AsyncOnRL` and overlapping experience collection and learning would likely be beneficial. Our implementation supports overlapping learning and collecting 1 rollout, but more overlap may be beneficial. The TP-SRL agent we build upon requires oracle knowledge, *e.g.* that the cabinet must be opened before picking.

We have presented Variable Experience Rollout (VER). VER combines the strengths of and blurs the line between `SyncOnRL` and `AsyncOnRL`. Its trains agents for embodied navigation tasks in Habitat 1.0 60%-100% faster (1.6x to 2x speedup) than DD-PPO with similar sample efficiency – saving 19.2 GPU-days on PointNav and 28 GPU-day for ObjectNav per seed in our experiments. On the recently introduced (and more challenging) embodied rearrangement tasks in Habitat 2.0, VER trains agents 150% faster than DD-PPO and is fast as SampleFactory (`AsyncOnRL`) on 1 GPU. On 8 GPUs, VER is 180% faster than DD-PPO and 70% faster than SampleFactory with better sample efficiency – saving 32 GPU-days vs. DD-PPO and 11.2 GPU-days vs. SampleFactory per skill in our experiments. We use VER to study rearrangement. We find the *emergence of navigation* in policies that ostensibly require no navigation when given access to navigation actions. This results in strong progress on Tidy House (+30% success) and shows that it may not always be advantageous to remove 'unneeded actions'.

**Acknowledgements.** The Georgia Tech effort was supported in part by NSF, ONR YIP, and ARO PECASE. EW is supported in part by an ARCS fellowship. The views and conclusions contained herein are those of the authors and should not be interpreted as necessarily representing the official policies or endorsements, either expressed or implied, of the U.S. Government, or any sponsor.

---

[3]Personal correspondence with authors.

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
