# OpenReview forum: "VER: Scaling On-Policy RL Leads to the Emergence of Navigation in Embodied Rearrangement"
_NeurIPS.cc/2022/Conference — NeurIPS 2022 Accept_

### Official Review · Reviewer_68WX · 2022-07-10

**Rating:** 6
**Confidence:** 2
**Soundness:** 2 fair
**Presentation:** 2 fair
**Contribution:** 3 good

**Summary:**

The paper focuses on the problems pertaining to SyncOnRL and AsyncOnRL and proposes a solution that combines the advantages of these methods. SyncOnRL suffers from the straggler effect, where the slowest worker decides the overall speed of execution, and the system capability is not utilized properly. AsyncOnRL solves this by taking action in an environment as soon as it is ready. Although this leads to efficient system utilization, it leads to reduced sample efficiency.
The proposed method, Variable Experience Rollout aims to mitigate these issues. The paper points out that collecting an equal number of time steps T per environment is easier to implement but is not a necessary requirement for reinforcement learning. VER collects a variable number of steps across the different environments. This overcomes staggler effect while at the same time is more sample efficient than AysnOnRL.

**Questions:**

Regarding Emergent Skills
* Would these skills not emerge when trained with AsyncOnRL on SyncOnRL?
* Is there something specific to VER that leads to the emergence of these skills? Or any method trained for a sufficient amount of time can lead to the emergence of these skills?


**Limitations:**

Limitations are not discussed properly.


**Strengths And Weaknesses:**

Strengths :
* The paper works on an exciting and vital problem of accelerating on-policy RL algorithms while maintaining sample efficiency.
* The idea is simple and original based on an ingenious observation about the training of the Reinforcement learning algorithm. It is a relatively novel idea.
* The amount of time required to train the RL algorithms when compared to the fastest SyncOnRL is much much less while maintaining the same sample efficiency. Making these algorithms more accessible to the general audience.
* The experimental details are comprehensive about the system capabilities, which is essential, especially for a paper that is mainly about utilizing system capabilities.
* Clear writing and neat plots.

Weaknesses
* Although the idea is exciting and novel, the presentation can be improved by including the following plot
    * it would be constructive to see one plot where :
       * x-axis: Policy training time (in hours or any other suitable time unit)
       * y-axis: Success Rate
       * Baselines: Fastest AsyncOnRL, Fastest SyncOnRL, VER
    * It is not clear whether VER leads to improvement in training time (ie the time required for the policy to converge to a high success rate).

* Emergent skills
    * Although it is an interesting result, it is not very clear how the results support the main idea of the paper. In my opinion, the analysis of these skills should be part of the appendix rather than the main paper.

---

> ### Author Response · Authors · 2022-08-02
> **Response**
>
> > the presentation can be improved by including the following plot (success-rate vs time)
>
> Agreed!
>
> We note this information (how quickly is X% success achieved as a function of wall-clock time) is implicitly present in the paper because we report throughput (steps per second) and learning curves (success vs steps), but we agree that it is useful to have this explicit plot.
>
> We have added this plot to Fig 4.
>
> > It is not clear whether VER leads to improvement in training time (ie the time required for the policy to converge to a high success rate).
>
> VER does lead to an improvement in training time. On 8 GPUs, VER has the best sample efficiency (either tied or out-right), Fig 3 and 4, and has the highest throughput, Tab 1 and Tab 3. Given that sample efficiency is at least as good and throughput is increased, training time must decrease.
>
> This point is even clearer with the new plot of success rate vs time. For a given OpenFridge success rate, VER achieves this in the least amount of wall-clock time (sometimes tied with baselines and sometimes outright).
>
> > Would these skills not emerge when trained with AsyncOnRL on SyncOnRL? Is there something specific to VER that leads to the emergence of these skills? Or any method trained for a sufficient amount of time can lead to the emergence of these skills?
>
> Reviewer 5YLN also asked this question, please see our response to them.
>
> > Emergent skills. Although it is an interesting result, it is not very clear how the results support the main idea of the paper. In my opinion, the analysis of these skills should be part of the appendix rather than the main paper.
>
> We are happy to move this to the appendix if other reviewers agree.

---

> > ### Comment · Reviewer_68WX · 2022-08-07
> > **Thanks for the response and some questions.**
> >
> > Thanks for the response and for adding the suggested graphs!
> > Could you add these graphs for 'PointNav' and 'ObjectNav' tasks? In my understanding, these seem to be the task where the VER training is beneficial (saving training days). In Fig. 4, all tasks take just hours to train, and although VER beats baselines in these environments, it would be helpful to see the 'success-rate vs time' graphs for tasks where VER can save days of GPU training time.

---

> > > ### Author Response · Authors · 2022-08-09
> > > **Response**
> > >
> > > > Could you add these graphs for 'PointNav' and 'ObjectNav' tasks?
> > >
> > > Happy to. We have added Figure A2 that shows Success vs. Time for these tasks. The result is consistent with the Habitat 2.0 tasks: VER reaches a given success threshold with significantly less wall-clock time. Specifically, to reach the maximum success achieved by DD-PPO (97.4% on PointNav and 13.0% on ObjectNav), VER uses 1.6x less compute on PointNav (saving 16 GPU-days) and 4.3x less compute on ObjectNav (saving 33.4 GPU-days).
> > >
> > > >  In my understanding, these seem to be the task where the VER training is beneficial (saving training days).
> > >
> > > Yes, VER does show significant training wall-clock time saving in Habitat 1.0 tasks (PointNav, ObjectNav; see above for details). However, we should note that these gains are not limited to these tasks. VER also saves _days of training time_ on Habitat 2.0 tasks (pick/place skills) with the full 500 million step budget needed for convergence. The reason why these speeds-ups may not be as stark in Fig 4 is that it uses 20 million steps to make a detailed study feasible — DD-PPO with 1 GPU would take over 1 month of wall-clock time to reach 500 million steps, making that experiment infeasible to run.

---

> > > > ### Comment · Reviewer_68WX · 2022-08-09
> > > > **Follow-up**
> > > >
> > > > Thanks for the reponse.
> > > > The perfomance gains of VER are much more clear now!

---

> > > > > ### Author Response · Authors · 2022-08-09
> > > > > **Thank you for your suggestions**
> > > > >
> > > > > Thank you for your suggestions, we agree that these plots have increased the clarity of our paper.

---

### Official Review · Reviewer_45yB · 2022-07-12

**Rating:** 3
**Confidence:** 4
**Soundness:** 2 fair
**Presentation:** 2 fair
**Contribution:** 2 fair

**Summary:**

This paper proposes a scaling technique for reinforcement learning (RL) to speed up sampling and learning in on-policy RL methods. The paper claims to blend the benefits of SyncOnRL and AsyncOnRL to increase both throughput and sample-efficiency. The main approach is to enable multiple environments to collect a variable number of steps simultaneously without synchronization. The paper performs experiments on several simulation scenarios and analyzes the performance of several baselines.

**Questions:**

+ The theoretical novelty and contribution of the paper are not clear. Why the proposed technical can theoretically improve both throughput and sample-efficiency?

+ The speed increase of VER on 8 GPU is only 70% faster than 1 GPU. Is the problem big enough to use the power of 8 GPU? Or is the algorithm for the 8 GPUs well parallelized? If 8 GPUs are not fully utilized, the motivation of using the proposed method on multiple GPUs seems not essential.

+ What is the difference between VER and AsyncOnRL? Theoretically/mathematically (not only quantitatively), why does VER perform better than AsyncOnRL?

+ The organization of the paper needs improvement. Experiment results are scattered throughout the paper, from introduction to the four sections of experiments. Some experiment setups also appear in the approach section. This increases the difficulty of understanding the theoretical novelty of the paper.

+ More details need to be provided to explain several concepts including TP-SRL and the architecture in Line 226.


**Limitations:**

No potential negative societal impact is perceived in the paper.

**Strengths And Weaknesses:**

+ The paper tries to address an important problem in RL to speed up sampling and learning.

+ The paper introduces an engineering scaling method that aims that takes advantages of SyncOnRL and AsyncOnRL.

---

> ### Author Response · Authors · 2022-08-02
> **Response (1/2)**
>
> > The theoretical novelty and contribution of the paper are not clear.
>
> As our paper describes, we present a simple technique for scaling on-policy batched on-policy RL systems and conduct rigorous evaluation. As reviewer 5YLN notices: “The focus of the paper is not on algorithmic novelty but rather on a highly performant RL system, which will be quite useful.”; and reviewer 68WX says: “The amount of time required to train the RL algorithms when compared to the fastest SyncOnRL is much much less while maintaining the same sample efficiency. Making these algorithms more accessible to the general audience.” There is no theoretical innovation in our work and our paper never claims any. We believe that a theoretical innovation is not a requirement to be a valuable contribution at NeurIPS.
>
> > Why the proposed technical can theoretically improve both throughput and sample-efficiency?
>
> Our paper does not claim that VER will always have better sample efficiency than SyncOnRL. We empirically find a case (ObjectNav) where it does improve sample efficiency and report that finding, but no general claims are made about sample efficiency.
>
> We do claim that VER will improve throughput over SyncOnRL for heterogeneous environments (where some environments take significantly longer to run than others). The reasons are explained in the intro and body of the paper, but are fairly straightforward and we are happy to summarize them again.
>
> The simple reason is the reduction of waiting time.
> The throughput of both VER and SyncOnRL is described mathematically as AmountOfExperienceCollected/(LearningTime + InferenceTime + SimulationTime + WaitingTime).
>
> Compared to SyncOnRL, VER reduces WaitingTime via its straggler-effect mitigations while all other terms remain the same, thus throughput will improve.
>
> See our answer below for how VER compares to AsyncOnRL.
>
> > What is the difference between VER and AsyncOnRL? Theoretically/mathematically (not only quantitatively), why does VER perform better than AsyncOnRL?
>
> There are two key differences between VER and AsyncOnRL that explain why VER performs better.
>
> The first is shown in Fig 1 -- AsyncOnRL overlaps experience collection with learning while VER does not. This explains why VER is more sample efficient. Due to this overlap, AsyncOnRL must learn with data collected from an older policy (L43-45). This effect is often referred to as policy lag and the data is often referred to as near-policy data. The on-policy objective used to optimize the policy is only well-defined for on-policy data and thus it follows that using near-policy data will reduce the efficiency of this objective. Methods like V-trace attempt to resolve this but they are only approximations. We are unaware of any work that proves that AsyncOnRL has reduced sample efficiency (and doing so is beyond the scope of our work), but this has been observed in prior work, Liu et al 2020, and observed in our work (Fig 4).
>
> The second difference is how multi-GPU scaling is achieved. VER uses the decentralized distributed method proposed in Wijmans et al 2020. In this method each GPU both collects experience and updates the model (see Sec 2.3 for more details). In AsyncOnRL framework we compare against, multi-GPU scaling is achieved by using additional GPUs for experience collection while learning is still performed on 1 GPU (explained in L291-L301). This difference explains why VER has better throughput on multiple GPUs.
>
> More formally, the maximum throughput of AsyncOnRL is the maximum number of samples per second the single GPU used for learning can process. This is a constant. As we increase the number of GPUs used for experience collection, we will approach and then reach this, but we cannot exceed it. The multi-GPU throughput of VER is nGPUs * ScalingFactor * VERSingleGPUThroughput.
>
> ScalingFactor and VERSingleGPUThroughput are constants, but nGPUs is not (it will have a maximum in practice, but theoretically it can be any non-negative value). Thus there must be a value of nGPUs such that
> nGPUs * ScalingFactor * VERSingleGPUThroughput > MaxAsyncOnRLThroughput
>
>
> > The speed increase of VER on 8 GPU is only 70% faster than 1 GPU.
>
> No, the reviewer is mistaken.
>
> As we say on L291, VER is 6.7x  faster on 8 GPUs than 1 GPU -- in Tab 3, 2861 (VER Mean, 8 GPUs) / 428 (VER Mean, 1 GPU) = 6.7x.
>
> The 70% faster result is VER on 8 GPUs vs SampleFactory on 8 GPUs -- L14-L15, in Tab 3 2861 (VER Mean, 8 GPUs) / 1662 (SampleFactory Mean, 8 GPUs) = 1.7x, which is 70% faster.

---

> ### Author Response · Authors · 2022-08-02
> **Response (2/2)**
>
> > More details need to be provided to explain several concepts including TP-SRL and the architecture in Line 226.
>
> As we describe in L220-223, we use TP-SRL as described in Szot et al 2021. This method decomposes GeoRearrange into a series of skills, Navigate, Pick, Place, Open {Fridge, Cabinet}, Close {Fridge, Cabinet}, and chains them together with a task planner (L221-223). The task planner is not learned and operates on privileged information (it knows if an object is inside/needs to be placed inside a container, i.e. the fridge). For a given HAB scenario, the task plan is the same across all instances and this is simply retrieved/used.
>
> While this information is all in Szot et al 2021, to make our work more self-contained, will include a section on TP-SRL in the supplement.
>
> We believe we have fully specified the skill policy architecture in L226-234 (note that the task planner has no architecture). If the reviewer could state what they believe is missing or not adequately explained we are happy to expand.

---

> ### Author Response · Authors · 2022-08-09
> **We hope that we have addressed your concerns**
>
> We hope that we have addressed your concerns. Are you satisfied with our response or do you have additional questions?

---

### Official Review · Reviewer_5YLN · 2022-07-15

**Rating:** 7
**Confidence:** 4
**Soundness:** 4 excellent
**Presentation:** 3 good
**Contribution:** 4 excellent

**Summary:**

This paper proposes an approach for bridging the gap between asynchronous and synchronous onpolicy RL to leverage the best of both worlds, preserving the throughput of the former and the  sample efficiency of the latter. They achieve this by parallely simulating N environments for different number of timesteps, thus avoiding being bottlenecked by the slowest environment. The authors evaluate their approach on habitat environments (including the more complex rearrange tasks from habitat 2.0), and empirically validate their claims.


**Questions:**

Questions -

-> Does the emergent navigation skill use described in section 6.2 also happen when using prior methods like DD-PPO, SampleFactory etc, even given more data (up to an order of magnitude more)?

-> Further analysis on the induced curriculum due to the relationship between simulation speed and difficulty of environments would be interesting. Does the proposed approach suffer in cases where the difficult environments are harder to simulate? How can this be mitigated?


**Limitations:**

Limitations are adequately discussed

**Strengths And Weaknesses:**

Significance

-> Performance in terms of sample efficiency and training speed are critical for RL agents. This paper provides an intuitively simple approach (i.e having a budget of samples for all environments each time before the policy update, as opposed to a fixed budget for each one of the environments) that improves on both metrics. This idea can be broadly adopted by the community due to its simplicity, and thus has potential to expedite iteration cycles for all RL researchers, pushing the field forward. Further the analysis performed by the authors comparing the sample efficiency and throughput to that of prior methods (pure async or pure sync) is quite thorough and exhaustive (including a comparison to a version of their method with all micro-optimizations, except the main idea - i.e having a fixed budget for all environments).

-> The authors include an interesting result that with access to skills not directly required for a task, the agent is able to leverage these to perform the task better given more data (Specifically, given access to navigation skills, the agent is able to perform pickplace better, even though navigation isn’t necessary for this task). It would be interesting to see if this emergent behavior can also be seen by running previous approaches (eg DD-PPO), even given more data. If prior methods require a lot more data to discover similar interesting emergent behavior, this would make the argument for adopting the proposed approach even stronger.

-> An interesting aspect of the approach is that it naturally induces a curriculum. Since easier environments are faster to simulate, samples from these are seen more often earlier on, and this helps learning. The authors mention this in passing, but I think there can be further analysis here, and perhaps a design change to specifically create a curriculum between easier / harder environments (difficulty might sometimes not vary exactly  with simulation speed, and I wonder if there are some trade-offs here that can be analyzed).


Quality
-> The empirical evaluation in terms of throughput and sample efficiency are the most critical components in establishing the author’s claims, and this is done quite exhaustively, even taking account variability in the number of GPUs available for training. Evaluation is included on simpler established navigation tasks from habitat1, and also on more complex longer horizon rearrangement tasks from habitat2 (which include multiple pick-place steps). The grasp action in these environments requires contact with the object and the robot to issue a command, as opposed to the ‘magic grasp’ commonly used in habitat environments (where the object is automatically grasped when close enough to the arm). This makes the control problem significantly harder, and brings the approach closer to real world settings (though that gap is still quite large, but addressing that is outside the scope of this paper).


Originality

-> The idea proposed is quite simple. The focus of the paper is not on algorithmic novelty but rather on a highly performant RL system, which will be quite useful. The related work adequately covers related work in the field, including relevant asynchronous and synchronous approaches.


Clarity

-> The paper is clearly written and motivated, and easy to follow. In order for this work to have an impact on the community the code must be released (since the focus of this work is a better RL system, and there are some micro-optimizations included that might be critical). The code is included in the supplementary, so I think the authors intend to release it.

---

> ### Author Response · Authors · 2022-08-02
> **Response**
>
> > Does the emergent navigation skill use described in section 6.2 also happen when using prior methods like DD-PPO, SampleFactory etc, even given more data (up to an order of magnitude more)?
>
> Yes, we believe so. The reason is while VER has significantly higher throughput (than DD-PPO and SampleFactory), the underlying core learning algorithm (PPO) is unchanged. However, we agree with the reviewer that the implicit curriculum in VER could give it a unique advantage.
>
> To empirically test this, we trained a Pick policy with DD-PPO for 500 million steps (the same as VER) and found it also exhibits emergent navigation (with some caveats). To quantify the amount of emergent navigation, we evaluated the Pick policy on NavPick. The Pick policy trained with VER gets 50% Success on NavPick while one trained with DD-PPO gets 42%. While this points to the possibility that VER leads to better emergent navigation skill usage, we note that the DD-PPO trained Pick policy performs ~6% worse pick, indicating that this may be due to worse training convergence. We did our hyper-parameter tuning with VER and it is entirely possible that VER and DD-PPO have slightly different optimal hyper-parameters that explains this difference. Overall, we are unable to definitively resolve this concern and will continue investigating this.
>
> > Does the proposed approach suffer in cases where the difficult environments are harder to simulate? How can this be mitigated?
>
> Great question. We have thought about this too.
>
> First, please note that the environments we studied for navigation do have the property that difficult environments are slower to simulate -- large houses are slower to render -- and we didn’t see a negative impact on training performance here. In fact, we found a small but measurable improvement on ObjectNav in the Matterport3D dataset.
>
> However, our intuition is aligned with the reviewer’s -- at some point there must be a negative effect. To test this, we performed a toy experiment where we artificially reduced the simulation speed of all environments except one by ~30x. Thus, nearly-all experience is collected from this one fast environment. As expected, the result is overfitting -- the agent performs well in that one single (fast) environment but does poorly in the vast majority of (slow) environments. The resulting Pick policy achieves 93% success when sampling training environments with the same frequency as training, but only 55% success when sampling the same environments uniformly.
>
> Ultimately, this pathological behavior is pointing to the underlying speed vs experience diversity trade-off. We can mitigate overfitting by forcing a minimum amount of experience from each environment. This would come at the cost of reduced throughput.
>
> We should note that AsyncOnRL is subject to the same trade-off. It too collects more experience from slower to simulate environments. So this trade-off isn’t unique to VER.
>
> > Further analysis on the induced curriculum due to the relationship between simulation speed and difficulty of environments would be interesting.
>
> Agreed! We think this is an interesting direction for future work.Thanks for the suggestion.
>
> > An interesting aspect of the approach is that it naturally induces a curriculum ... and perhaps a design change to specifically create a curriculum between easier / harder environments.
>
> Great idea! We developed VER for its system benefits and then empirically observed this induced curriculum. One could use VER explicitly for a curriculum and possibly see some system benefits instead. While studying this is beyond the scope of our work, we will note these points as directions for future work. Thanks again.
>
>
> >  In order for this work to have an impact on the community the code must be released
>
> Agreed! We think that releasing code is paramount to the success of this work and we will not be waiting for publication acceptance to do so. We have already updated our code-base to maintain parity with habitat-lab/sim main and have begun the process of publicly releasing code.

---

> > ### Comment · Reviewer_5YLN · 2022-08-10
> > **Rebuttal Response**
> >
> > I thank the authors for their response, I will be maintaining my score in favor of acceptance.

---

### Official Review · Reviewer_ep2X · 2022-07-16

**Rating:** 5
**Confidence:** 3
**Soundness:** 3 good
**Presentation:** 2 fair
**Contribution:** 3 good

**Summary:**

The authors present VER, a method that could both tackle the challenge in synchronous RL that the policy updates need to wait for all steps done in all environments (straggler effect), and the challenge in asynchronous RL that we need to explicitly tackle stale samples when do on-policy learning. Specifically, VER is an environment-dependent variable step size training method that uses variable step numbers according to environments. Evaluation experiments are done sufficiently.

**Questions:**

Please check the 2 points stated above in the weakness section.

**Strengths And Weaknesses:**

Strengths
---------------------------------
This paper discusses an important problem in the RL community, i.e., what techniques should we use in the context that we usually vectorized a batch of environments in simulations to speed up the training time. Overall, the reviewer enjoys reading the author's analysis of problems in AsyncOnRL and SyncOnRL. And the motivation for variable step size rollout makes sense to me. Experiments are sufficient, and efforts are obvious.

Weakness
---------------------------------
- (1) The authors fail to describe how exactly the step length is decided. As a comparison, the baseline DD-PPO [1] paper clearly describes the whole training solution.
 - (2) Although several tasks are evaluated to test the efficiency of VER, it's hard for the reviewer to determine the claimed performance, for example, "in Habitat 2.0 VER is 150% faster on 1 GPU and 200% faster on 8 GPUs". At least VER's performance gain is not obvious in Fig. 4.

Ref:
-----------------
- [1] Wijmans E, Kadian A, Morcos A, et al. Dd-ppo: Learning near-perfect pointgoal navigators from 2.5 billion frames[J]. arXiv preprint arXiv:1911.00357, 2019.

---

> ### Author Response · Authors · 2022-08-02
> **Response**
>
> > The authors fail to describe how exactly the step length is decided
>
> We are unsure what the reviewer means by “step length” as this phrase is never used in our paper, but will try to answer the question based on our best interpretation. If the reviewer can clarify what they meant, we are happy to update our answer.
>
> If by “step length” the reviewer meant the number of steps collected from each environment, this is described in L123-128. The core idea of our algorithm (VER) is that we do not need to explicitly decide how many steps to collect per environment. Instead, this is decided implicitly based on the runtime of an environment. VER collects experience from each environment as quickly as the environments generate them and VER stops collecting experience once T*N total steps have been collected. The number of steps collected per environment is then just how many we collected in that time frame. There is no explicit “step length” parameter.
>
> > Although several tasks are evaluated to test the efficiency of VER, it's hard for the reviewer to determine the claimed performance, for example, "in Habitat 2.0 VER is 150% faster on 1 GPU and 200% faster on 8 GPUs". At least VER's performance gain is not obvious in Fig. 4.
>
> VER’s performance gains (speed-ups) are not visible in Fig 4 because that is not what Fig 4 shows. As the Fig 4 caption and Section 5.2 describe, Fig 4 shows sample efficiency results (accuracy vs #steps), not compute speed-ups.
>
> As described in Section 5.1 L270-281 and the Table 3 caption, the speed-up claims are supported by results in Table 3. Here are the relevant speeds from that table: 428 (VER Mean, 1 GPU) / 174 (DD-PPO Mean, 1 GPU) = 2.5, which is a 150% speed-up; 2861 (VER Mean, 8 GPUs) / 1066 (DD-PPO Mean, 8 GPUs) = 2.7, which was rounded up to a 200% speed-up.
>
> At the request of Reviewer 68WX have added Figure 4 (Lower) that shows a combined view of throughput and sample efficiency, which may be helpful.
>
> Overall, we hope that our answers have addressed the reviewer’s two stated concerns and that they will increase their rating.

---

> > ### Comment · Reviewer_ep2X · 2022-08-08
> > **My misunderstanding of Fig. 4 has been cleared, but still have problem with the clarity of VER**
> >
> > Thanks for the authors' kind reply. Yes, I have a misunderstanding of Fig. 4, and now it's clear.
> >
> > **About empirical results supporting the speed-up claims:**
> >
> > - Now it's clear that Table 2 (you wrongly typed Table 3 in your response) includes the main empirical results supporting the speed-up claims. The SPS results show clear performance gains.
> > - **Additional concern**: since it's based on only the open-fridge task since it's quite a challenging one, I am wondering, did you try experiments on other challenging tasks too? This is important since, for empirical evaluations, comprehensive benchmark results will better support the claimed performance gain.
> >
> > **About the clarity of VER method**
> >
> > Thanks for the authors' kind explanation about how variable environment steps are achieved. I also read the other reviewer's comments, I think the clarity concern of VER method still applies here.
> >
> > - Three Reviewers, including 45yB, 68WX, and me rated the presentation of this paper as "fair".
> > - As a comparison, Dd-ppo has a better presentation of their methods.
> > - Would the authors consider better organizing this paper for readers from general RL background?
> >
> > Since I've known how VER works and table 2 supports their claimed speed-up performance gains, I will raise my score a little bit.
> >
> > To the Area chair, please consider my two concerns here when making decisions.

---

> > > ### Author Response · Authors · 2022-08-09
> > > **Response**
> > >
> > > We are glad the misunderstanding around speeds-up has been resolved. Thank you for updating your views and the score! We answer the new questions asked below (which is again a misunderstanding around what is already present in the paper)
> > >
> > > > Now it's clear that Table 2 (you wrongly typed Table 3 in your response) includes the main empirical results supporting the speed-up claims.
> > >
> > > Sorry for the confusion. We used the review version pdf for line numbers, table numbers, etc in our response to indicate that this material was present in the document before the rebuttal. Starting from this response, we will switch to using numbers in the current version to avoid further confusion, but note that unless stated this information was present in the submitted review version.
> > >
> > > Also, please note that Table 2 is not our only empirical result (more details below).
> > >
> > > > since it's based on only the open-fridge task since it's quite a challenging one, I am wondering, did you try experiments on other challenging tasks too?
> > >
> > > The results in Table 2 are on the open-fridge task, but those are not the only experiments or results in the paper.
> > >
> > > As we describe in the abstract (L11-L13), introduction (L66-L70), Section 3 (L175-L210), and Table 1 caption, we also conducted large-scale and rigorous evaluations on navigation tasks in Habitat 1.0 (PoingNav (Anderson et al, 2018), and ObjectNav (Batra et al, 2020b)).
> > >
> > > As described in L187-L205, we see consistent and similar throughput improvements. VER is 1.6x faster than DD-PPO when training agents for PointNav and 2x faster for ObjectNav.
> > >
> > > ObjectNav on the Matterport3D dataset is a challenge for learning systems because the environments have large differences in size and therefore rendering speed (~5x) and ‘resetting’ the environment can be very expensive (on the order of seconds compared to on the order of milliseconds for a step). These are two system challenges that we designed VER to mitigate and its mitigations are effective in this setting.
> > >
> > > Finally, for the rebuttal, we have now added Figure A2 that shows PointNav/ObjectNav success vs. time. It shows VER reaches a given success threshold with significantly less wall-clock time. Specifically, to reach the maximum success achieved by DD-PPO (97.4% on PointNav and 13.0% on ObjectNav), VER uses 1.6x less compute on PointNav (saving 16 GPU-days) and 4.3x less compute on ObjectNav (saving 33.4 GPU-days).
> > >
> > > We would like to note two things: First, that the existing benchmarking results in Tab 1 and Tab 2 are the result of 1.3 years of GPU time, so this experimental analysis is rigorous and large-scale. Second, while we only provided detailed system comparisons for one Habitat 2.0 skill, OpenFridge, we have used VER to train all the skill policies. The accuracy of such a skill-chaining system is provided in Section 6.1 (L306-L316), and the high-level system trends are similar.
> > >
> > > >  Would the authors consider better organizing this paper for readers from general RL background?
> > >
> > > Happy to.
> > >
> > > Our organization largely follows systems papers published in similar venues: DD-PPO (ICLR ‘20), SampleFactory (ICML ‘21), SEED-RL (ICLR ‘20), and HTS-RL (NeurIPS ‘20) -- describing the system, describing the task, describing system results, and describing task results.
> > > Since we study two largely distinct task settings and study them in different depths (Habitat 1.0 Navigation vs. Habitat 2.0 Rearrangement), we chose to present these independently as we felt that that made the paper clearer.
> > >
> > > Does the reviewer have specific suggestions for improvement?
> > >
> > > > Since I've known how VER works and table 2 supports their claimed speed-up performance gains, I will raise my score a little bit.
> > >
> > > We thank the reviewer for this increase. We hope that with the additional support of Table 1 and the two experiments in the paper that appear to have been missed, the reviewer will increase their score again because we don’t believe our evaluation is limited.

---

### Author Response · Authors · 2022-08-02
**Response to Reviewers**

We thank the reviewers for their comments and feedback. We are pleased that our reviewers found the follow:

- That our work “discusses an important problem in the RL community” (ep2x), “can be broadly adopted by the community due to its simplicity, and thus has potential to expedite iteration cycles for all RL researchers, pushing the field forward” (5YLN), “simple and original based on an ingenious observation about the training of the Reinforcement learning algorithm” (68WX), and “introduces an engineering scaling method that aims that takes advantages of SyncOnRL and AsyncOnRL” (45yB).
- That our experiments are “sufficient” (ep2X), our “experimental details are comprehensive about the system capabilities” (68WX), and our analysis to be “quite thorough and exhaustive” (5YLN).
- That our emergent navigation result is “an interesting result” (5YLN).
- That our paper is “clearly written and motivated, and easy to follow” (5YLN), with  “clear writing and neat plots” (68WX), and that “the reviewer enjoys reading the author's analysis of problems in AsyncOnRL and SyncOnRL” (ep2X).

We will address questions and concerns individually below.

---

### Meta-Review · Area_Chair_UmEp · 2022-08-28

**Recommendation:** Accept
**Confidence:** Less certain

**Metareview:**

The paper proposes a novel method that takes the best of both worlds: synchronous and asynchronous on-policy RL methods.
The rebuttal nicely addressed the concerns of most reviewers.
Why the method makes sense and has benefits is rather straightforward and intuitive (which is a good thing!). The paper is clearly an experimental paper / systems paper with very extensive evaluations that show significant practical benefits. The method potentially has large practical impact, which is an important contribution to the community (and a valid - though less common - NeurIPS paper format). Hence I disagree with reviewer 45yB about requiring a theoretical novelty/contribution.


*** not visible to authors as posted after discussion phase ***

Agree to raise the score

NeurIPS 2022 Conference Paper1458 Reviewer ep2X

18 Aug 2022

Excellent work, impressive results!

I will raise my score.


**Award:**

No

---

### Decision · Program_Chairs · 2022-09-14

Accept